# Optimizing the Path of Plug Tray Seedling Transplanting by Using the Improved A* Algorithm

**Xiaojun Li** [1], **Weibing Wang** [1,*], **Ganghui Liu** [1], **Runze Li** [1] and **Fei Li** [2]

1   College of Mechanical and Electrical Engineering, Shihezi University, Shihezi 832003, China
2   College of Mechanical Engineering and Automation, Beihang University, Beijing 100083, China
*   Correspondence: wwb_mac@shzu.edu.cn

**Abstract:** In greenhouse nurseries, one of the important tasks of the automatic transplanter is replanting missing or bad seedling holes with healthy seedlings. This requires the transplanter to spend significant time moving between the supply trays and target trays during replanting. The diversity and complexity of the transplanting routes affect transplanter efficiency. Path planning method can find a better path for the manipulator and improve the efficiency of transplantation. The A* algorithm (A*), which is one of the optimal path search algorithms, is often used in practical applications of path planning. In this paper, the heuristic function of the A* is optimized by the ant colony algorithm (ACA), and an improved A* algorithm (Imp-A*) is obtained. Simulation tests and transplanting trials of Imp-A*, A*, ACA, Dijkstra (DA), and common sequence method (CSM) were carried out using 32-, 50-, 72-, and 128-hole plug trays. The results show that Imp-A* inherits the advantages of A* and ACA in terms of path planning length and computation time. Compared to A*, ACA, DA, and CSM, the transplanting time for Imp-A* was reduced by 2.4%, 12.84%, 11.63%, and 14.27%, respectively. In just six trays of transplanting tasks, Imp-A* saves 60.91 s compared to CSM, with an average time saving of 10.15 s per tray. The combination optimization algorithm has similar application prospects in agriculture.

**Keywords:** greenhouse; replanting; seedlings; path planning; optimization algorithms



## 1. Introduction

Seedling transplanting has the advantage of climate compensation, which can improve the survival rate and output. It has replaced the traditional direct seeding method in the North-West of China [1–4]. Seedling leakage, seedling germination, and seedling stunting occur in the cultivation of seedlings, with healthy seedlings accounting for 80–95% of the population [5]. To ensure the consistency and effectiveness of field transplanting, seedling growth must be monitored, and seedlings missing, or bad seedling holes should be eliminated and replaced with healthy seedlings. At present, the common sequence method (CSM) is still used for the replanting of seedlings, but this method is invalid in path planning [6].

The researchers used machine vision technology [6–9] and image processing algorithms [10–12] to detect the growth status of plug tray seedlings and identify unhealthy seedlings in preparation for replanting. When the replanting of seedlings begins, the manipulator needs to grasp the healthy seedlings from the plug tray that supplies seedlings and transplants them to the plug tray that needs replenishment seedlings, and this process involves the reciprocating movements of the manipulator with different routes. In the case of manipulator moving speed, the shorter the moving route, the higher the transplanting efficiency. Seedling transplanting path planning is similar to the traveling salesman problem (TSP) [13,14], where the optimal path is found after passing through all the city nodes entering from the source. The length of the transplanting path and computation time are two important indexes to measure the algorithm model. Greedy algorithm (GRA)

and genetic algorithm (GA) are used to solve single-source path planning. According to the characteristics of this algorithm, GRA looks for local optimal solutions step by step, the final result may be infinitely close to the optimal solution, but it requires more time than other algorithms. GA has advantages in computation time, but it does not always find the optimal solution. The greedy genetic algorithm (GGA), developed by combining GRA and GA, has been used to good effect for the thinning and transplanting of plug tray seedlings, with improved path optimization and computation time [15–21]. Ant colony algorithm (ACA) is a common algorithm to solve combinatorial optimization problems, and its optimal path is determined by the pheromone concentration released by bionic ants. However, the biggest drawback of the ACA is that it tends to fall into local optimality and converge too slowly. Resulting ACA does not perform well in terms of path planning length and algorithm computation time, but improved and optimized ACA often shows excellent performance [13,22–26].

To improve the computing speed of the path planning algorithm model, researchers have begun to explore more effective algorithm models. A* algorithm (A*), as one of the optimal path search algorithms, adopts the breadth-first search strategy, a heuristic function to guide the search direction, effectively shortens the path search length computation time. With the development of computer science, people's demands and expectations of the A* algorithm have grown more and more, and they have started to search for faster and more efficient ways to improve the A* algorithm. The combination of ACA and A* as a new method, by combining bidirectional search with intelligent ACA, the selection factor of the heuristic function of A* is obtained, then use the factor to improve the algorithm function and get a more efficient improved A* algorithm (Imp-A*) [27,28]. However, there are relatively few cases in which Imp-A* is applied to transplanter path planning.

The calculation time and path optimization of the seedling transplanter path planning algorithm model is studied. Under the guidance and optimization of the heuristic function of ACA, this method is applied to seedling transplanting path planning of transplanters based on the network grid model. The Imp-A* searches from source O to target grid node E and also searches from target grid node E to source O. The algorithm terminated when the same coincidence grid node is searched in both directions. By training the heuristic function of A* with ACA, more accurate heuristic function factors can be obtained in a shorter time, and the efficiency of A* can be improved. The main contributions of this study were as follows:

1.  The A* algorithm was optimized and improved in combination with the ACA, an improved A* algorithm (Imp-A*) was obtained, and the algorithm model was applied to the field of plug tray seedling transplantation successfully, and it provided the optimal path for replanting seedlings of the manipulator.
2.  The path planning length and calculation time data of the Imp-A* model were obtained through simulation tests, and compared with simulation data of other algorithm models, the optimal route length and calculation time of the algorithm model were obtained.
3.  Based on the simulation model, the transplanting trials of replanting were designed, the Imp-A* in this paper was applied to the practical operation, and the robustness of the algorithm was tested; the efficiency and practicability of the Imp-A* were verified.

The rest of the paper is divided into the following sections: The second section introduces the working principle of transplanter and several path planning algorithm models. In the third section, the simulation test and transplanting trials results are explained and analyzed in detail. Finally, Sections 4 and 5 describe the discussion and introduction of this study.

## 2. Materials and Methods

### 2.1. Structure of the Device and Path Planning Principles

Figure 1a shows the three-dof transplanting device supporting X, Y, and Z movements. As shown in Figure 1b, dark circles represent healthy seedlings, and white circles represent

holes removed from poor-quality seedlings. When transplanting, the manipulator starts at the O-source point and transplants healthy seedlings from the seedling supply trays (referred to as S tray hereafter) to the inferior seedling holes in the target trays (referred to as T tray hereafter).

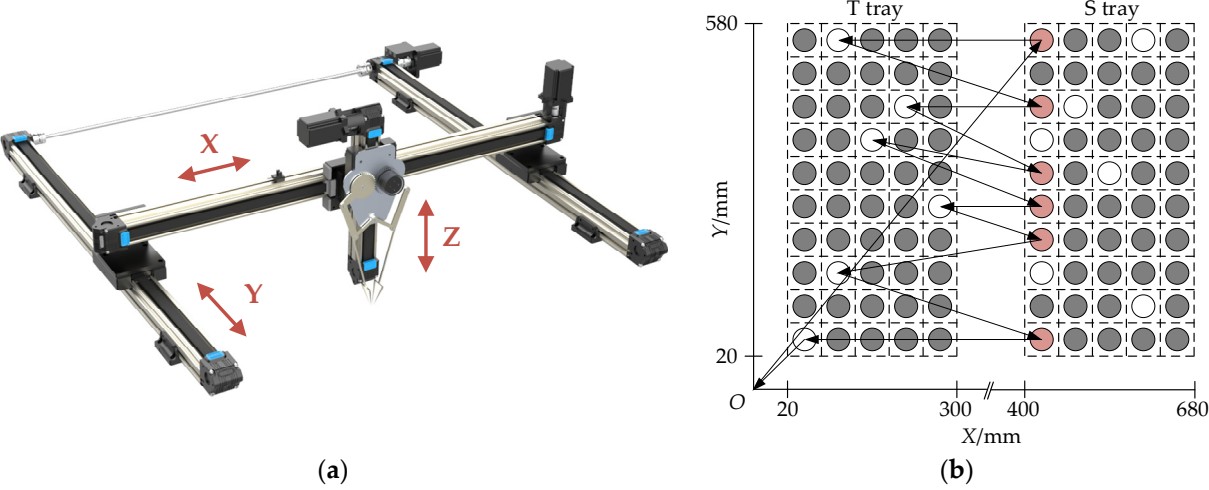

**Figure 1.** Diagram of (**a**) the 3-dof transplanting device and (**b**) tray location and transplanting path. The red circles indicate healthy seedlings to be transplanted; the grey circles indicate healthy seedlings; the white circles indicate empty holes.

The random position of the holes in the S tray for healthy seedlings and the T tray for poor quality seedlings determines the variety and complexity of the transplanting path planning. For a given control system parameter, the three-dof transplanting system moves at a given speed, and the transplanting unit needs to be paused during transplanting, depending on the speed at which the transplanting task is completed. The transplanting path optimization is based on the traditional Common Sequence Method (CSM) and uses a path search algorithm to plan the shortest possible reciprocal movement sequence for the manipulator, shortening the length of its movement route to reduce the pause time of the transporter and improve the efficiency of the transplanter.

*2.2. Path Planning Methods*

The seedling trays with different numbers have the same external size (560 mm in length and 280 mm in width). A total of 32 (specification: 4 × 8), 50 (specification: 5 × 10), 72 (specification: 6 × 12), and 128 (specification: 8 × 16) holes are frequently used in the North-West of China. It is known from reference [21] that the randomness of the positions of M ($1 \leq M \leq G$) holes in the S tray and N ($M \leq N \leq G$) healthy seedlings in the T tray determine that there are paths available for replanting seedlings. In the case of 50 holes, if inferior quality seedlings account for 5–20% of the total, M = 5 in the S tray and the healthy seedlings N = 45 in the S tray, there are $45! \times 5!/(45-5)! \approx 1.76 \times 10^{10}$ alternative transplanting paths. If the traversal method is used to find the optimal path, the calculation is too much, and the operation time is too long, it is not in line with the concept of fast and efficient automatic transplanting equipment. Therefore, it is necessary to find an effective path planning algorithm.

As shown in Figure 2a, the location of the seedless holes in visual image processing is transmitted to PC in the form of a digital matrix that is converted to a grid map of coordinates when a path planning model is established. Set the S tray numeric matrix to S and the T tray numeric matrix to T, scanning the matrix from left to right, top to bottom, corresponding to the coordinate grid, as shown in Figure 2b. The position of the holes in the S tray can be expressed in $S_i$, and the position of holes in the T tray can be expressed in $T_j$, corresponding position of the holes of missing and bad seedlings.

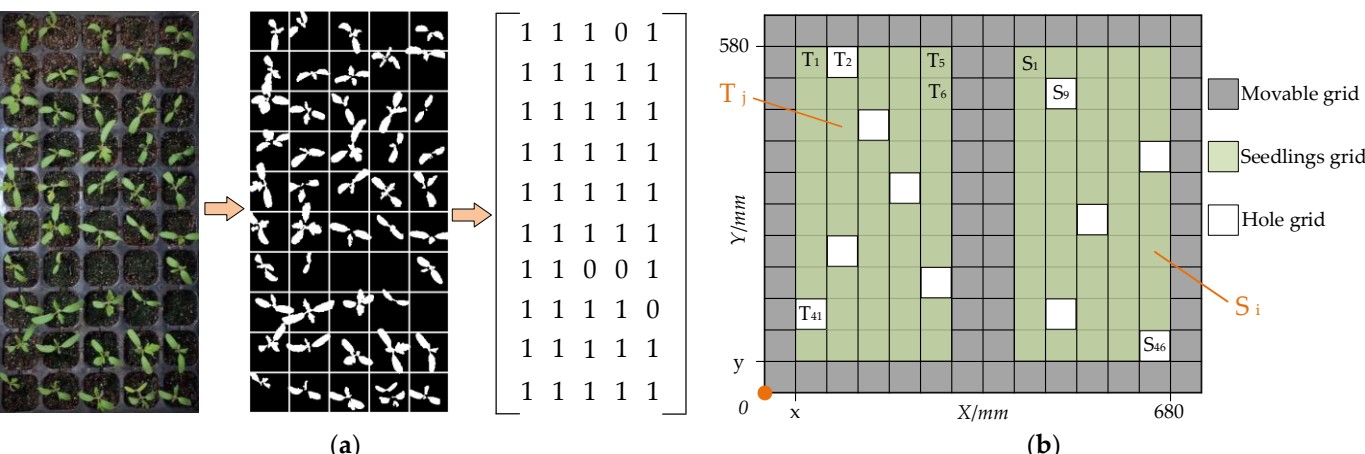

**Figure 2.** Diagram of (**a**) digital matrix of plug tray information and (**b**) grid network model. $S_i$ indicates the hole location of the seedling supply tray and $T_i$ indicates the hole location of the target tray.

The length of path planning is the sum of the horizontal and vertical distances. For greater clarity, the path is represented by the shortest distance between two points. Starting with the coordinates of source point O, the manipulator moves toward the X and Y axes. First, the manipulator reaches the S tray and selects a healthy seedling to transplant into the hole in the T tray. The distance traveled by the manipulator for a single replanting task can be expressed as:

$$\left(x_{S_i} - x_{T_j}\right) + \left|y_{S_i} - y_{T_j}\right| + \left(x_{S_i} - x_{T_j'}\right) + \left|y_{T_j'} - y_{S_i}\right| \tag{1}$$

where x and y are the horizontal and vertical coordinates of the holes position of the manipulator, $x_{S_i}$, and are the horizontal and vertical coordinates of healthy seedling holes located in the S tray to be captured in the next step, $x_{T_j'}$, and are the horizontal and vertical coordinate of the holes location of the T tray to be transplanted into seedlings.

Where $x_{T_j}$ and $y_{T_j}$ are the horizontal and vertical coordinates of the position of the cavity in which the transplanter was located at the time of the previous transplanting step, and $x_{S_i}$ and $y_{S_i}$ are the horizontal and vertical coordinates of the position of the healthy cavity in the supply cavity to be picked up in the next step, and $x_{T_j'}$ and $y_{T_j'}$ are the horizontal and vertical coordinates of the empty holes in the destination holes to be replanted.

### 2.2.1. Common Sequence Method (CSM)

CSM is one of the most commonly used transplanting methods in practical production. The classic method is to scan the S and T tray from top to bottom, from left to right, and remove healthy seedlings from the S Tray, transplanting healthy seedlings into the T tray. This method does not plan or compare the path of plug seedlings nor involve the operation of a path planning algorithm. There is no algorithm for calculating time because the path planning of the process has been determined. Therefore, the CSM only takes part in the comparison test of the path planning length and does not take part in the comparison test of the calculation time (Figure 3).

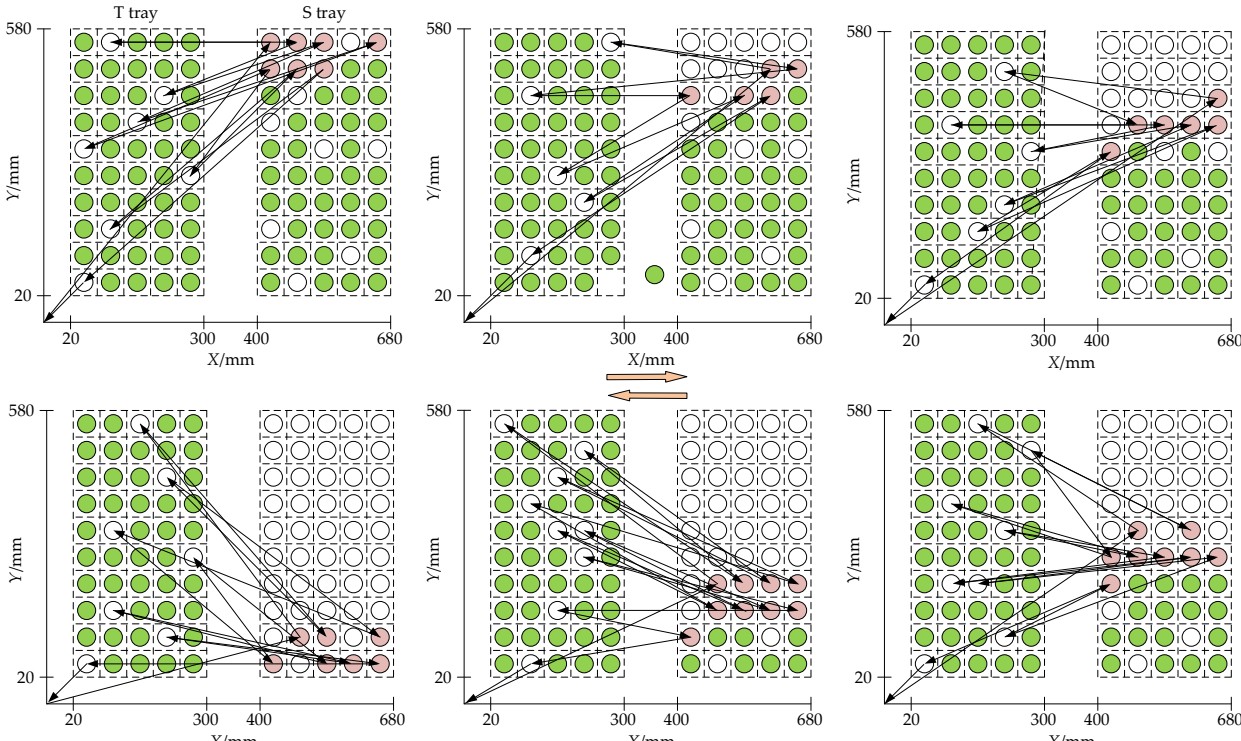

**Figure 3.** Diagram of CSM to complete 50 holes transplanting path planning. The red circles indicate healthy seedlings to be transplanted; the green circles indicate healthy seedlings; the white circles indicate empty holes.

### 2.2.2. Dijkstra Algorithm (DA)

DA is a greedy algorithm that solves the problem of the shortest path for a single source by first finding the shortest path, then the second shortest path is found by referring to this shortest path until the shortest path from the source to the target node is found. DA was applied to pot seedling transplanting path planning, which could be transformed into the scanning of the S tray and T tray from bottom to top and from left to right; starting from the source point, the manipulator grasps the healthy pot seedlings in the S tray nearest to itself and transplants them to the first hole position scanned in the T tray. Then, using this location as a source of greed, look for the nearest healthy pot seedlings in the S tray and transplant them to the next hole scanned in the T tray (each node can be visited only once). As shown in Figure 4, circulate until all the holes in the T tray are filled, and transplanting all healthy pot seedlings into the S tray was considered an experimental group. Several groups of experiments were carried out, and the calculation time required by the DA model to complete the path planning and the path length of transplanting were counted.

### 2.2.3. Ant Colony Algorithm (ACA)

ACA means that the feasible solution of the problem to be optimized is expressed by the path of ants, and all the paths of the ant colony constitute the solution space of the problem to be optimized. Ants with shorter paths release more pheromones, and over time, the accumulation of pheromones increases in ants with shorter paths, and the number of ants choosing path increases [25,27]. Finally, the ant colony focuses on the optimal path under the action of positive feedback, and the corresponding path is the optimal solution to the problem to be optimized.

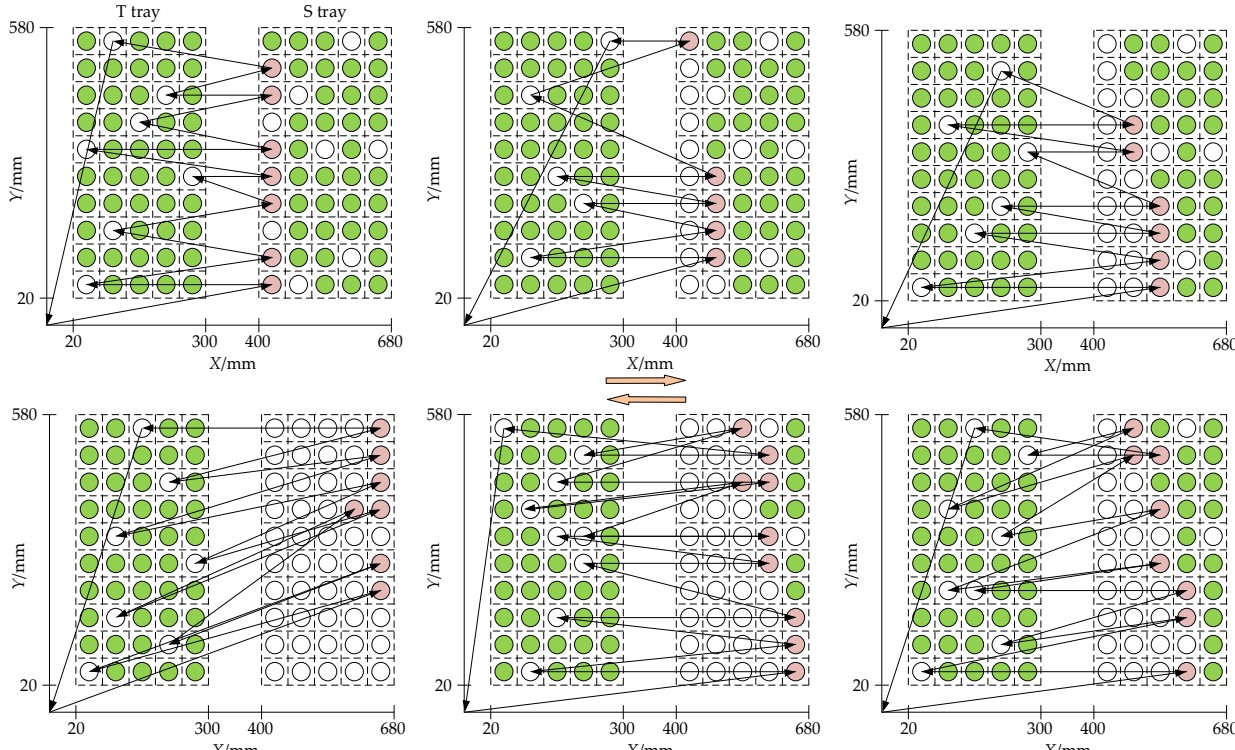

**Figure 4.** Diagram of DA to complete 50 holes transplanting path planning. The red circles indicate healthy seedlings to be transplanted; the green circles indicate healthy seedlings; the white circles indicate empty holes.

To ensure that each healthy seedling grid in the S tray and each empty grid in the T tray can be visited only once, Openlist and Closelist were created to store the grids that need to be visited, and the grids were visited by ants. Closelist allows ants to retrace their journeys after completing a path search. Assuming that the n ant set U remains the same, and in U, the starting point O for all ants is used as the starting point U. Setting the rules for each ant's movement: from the starting point to the healthy seedling hole in the S tray, the hole will be added to the Closelist when they leave, then the ant moves to the hole position in T tray, and when they leave, the hole will be added to the Openlist. The ant must then maintain a healthy seedling position in S tray and circulate until it visits all the holes. Any ant k will make a probabilistic decision when implementing these rules. After selecting grid i, it visits the next grid j with the following probabilities:

$$
P_{ij}^k(t) = \begin{cases} \dfrac{[\tau_{ij}(t)]^{\alpha} \cdot (\eta_{ij})^{\beta}}{\sum_{j \in N_j^k} [\tau_{ij}(t)]^{\alpha} \cdot (\eta_{ij})^{\beta}} & , \text{if } i \in N_j^k \\ 0 & , \text{else} \end{cases}
\tag{2}
$$

When the ant is in j, it visits the next grid i with the following probabilities:

$$
P_{ij}^k(t) = \begin{cases} \dfrac{[\tau_{ij}(t)]^{\alpha} \cdot (\eta_{ij})^{\beta}}{\sum_{i \in N_i^k} [\tau_{ij}(t)]^{\alpha} \cdot (\eta_{ij})^{\beta}} & , \text{if } j \in N_i^k \\ 0 & , \text{else} \end{cases}
\tag{3}
$$

where $\tau_{ij}(t)$ is the pheromone between grid i and j after t iterations, $\eta = 1/d_{ij}$ is the heuristic information, and $d_{ij}$ is the distance from grid i to grid j. $\alpha$ is the degree effect of pheromone concentration on ant behavior, $\beta$ is the degree effect of path length on ant behavior, $N_i^k$ denotes an unvisited seeded hole in grid S, and $N_j^k$ denotes an unvisited empty hole in the grid T.

When n ants pass through all the holes in the grid T, the pheromones in their path are with the Equations (4) and (5) as follows:

$$\tau_{ij}(t+1) = (1-\rho)\tau_{ij}(t) + \Delta\tau_{ij}(t,t+1) \tag{4}$$

$$\Delta\tau_{ij}(t,t+1) = \sum_{k-1}^{m} \Delta\tau_{ij}^{k}(t,t+1) \tag{5}$$

where $\rho$ is the fluctuation of pheromones ($\rho \in (0,1)$ ), $\Delta\tau_{ij}^{k}(t,t+1)$ is the total amount of pheromones released by ant k on the path of the current cycle (i, j). $\Delta\tau_{ij}(t,t+1)$ denotes the pheromone increment on the path (i, j) in each cycle. Pheromones can be updated by various methods, and volatilization is one way of pheromone update; when the path is not selected by ants, the pheromone on that path will also volatilize over time, which leads to an infinite accumulation of pheromones that mislead ants. $\Delta\tau_{ij}^{k}(t,t+1)$ can be defined as follows:

$$\Delta\tau_{ij}^{k}(t,t+1) = \begin{cases} Q/L^{k} & , \text{path } (i,j) \text{ is visited in t iterations} \\ 0 & , \text{else} \end{cases} \tag{6}$$

where Q is a constant representing the number of pheromones stored in a given ant path search after an iteration, $L^{k}$ being the total length of the ant k path search.

ACA flowchart, shown in Figures 5 and 6a, is the ACA's roadmap for planning a 50-hole plantings path. Set the maximum number of iterations per path search performed by the ACA to 50, and the variation between the optimal path length and average path length of the path optimization task segment and the number of iterations performed by the ACA model is shown in Figure 6b. Experimental results show that the algorithm is effective and has good convergence. Thus, the relatively optimal transplanting path is the route chosen by the vast majority of ants.

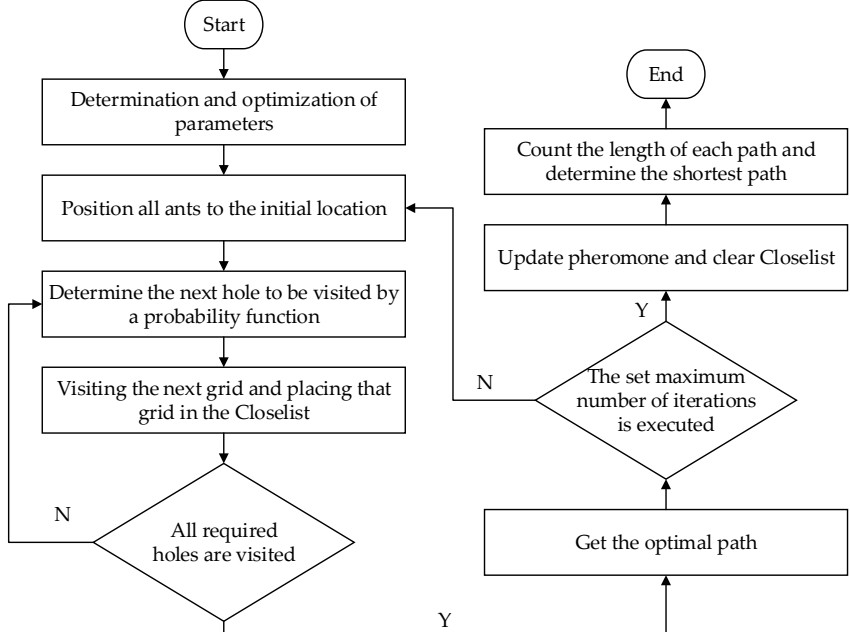

**Figure 5.** Flowchart of ACA.

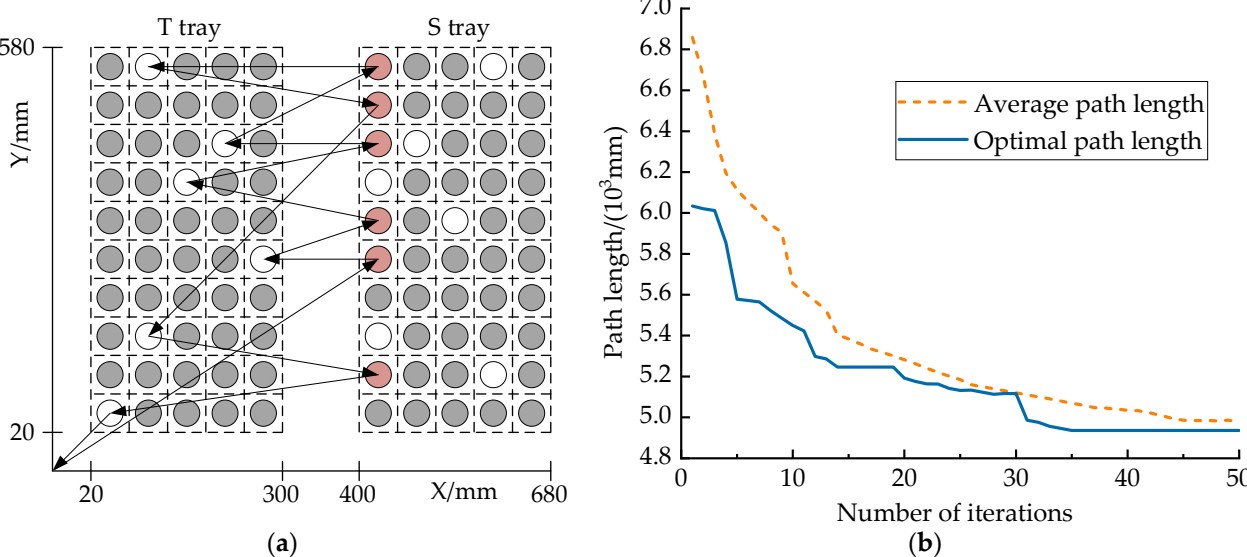

**Figure 6.** Diagram of (**a**) path planning of ACA and (**b**) variation of ACA path planning length with the number of iterations. The red circles indicate healthy seedlings to be transplanted; the grey circles indicate healthy seedlings; the white circles indicate empty holes.

### 2.2.4. A* Algorithm (A*)

A* is one of the most popular and effective heuristic search path optimization algorithms. The innovation of this algorithm is to select the next node using the known global information and carry out probability analysis and cost analysis [28]. According to the estimated distance between the current node and the target node, the algorithm evaluates the nodes on the optimal path, realizes the search of the nearest node, and improves search efficiency.

The heart of A* is the creation of the heuristic function. Assuming that $\varepsilon$ is the current grid node position, the heuristic function for this node can be expressed as:

$$f'(\varepsilon) = g(\varepsilon) + h'(\varepsilon) \tag{7}$$

where $g(\varepsilon)$ is the actual distance function from the source point O to the current grid (S grid) node $\varepsilon$, and $h'(\varepsilon)$ is the minimum estimated distance from the current grid node $\varepsilon$ to the target grid(T grid) node e. Since $h'(\varepsilon)$ is an estimate of the minimum distance ignoring obstacles and movement rules, $h'(\varepsilon)$ is defined as the Euclidean distance from the S grid node $\varepsilon$ to the T grid e in the path-finding model; if we make h = 0, A* will degrade to DA.

Suppose the source point coordinates are $O(O_x, O_y)$, a healthy seedling hole coordinates $(S_{ix}, S_{iy})$ in the S grid is the intermediate node, and empty hole coordinates $(T_{jx}, T_{jy})$ in the T grid is the target node, then the heuristic function can be expressed as:

$$f'(\varepsilon) = g(\varepsilon) + \sqrt{\left(S_{ix} - T_{jx}\right)^2 + \left(S_{iy} - T_{jy}\right)^2} \tag{8}$$

where $g(\varepsilon) = \sum\limits_{t=1}^{n} d(t)$, $d(t)$ is the distance between two neighboring grids.

Instead of traversing all nodes, A* introduces heuristic information to guide the movement to the target node, making it easier for the algorithm to accelerate computations. However, ignoring a large number of "dotted points" in the search process can affect computations. Because of the complexity of the real environment, it is sometimes wrong to introduce heuristic information into the cost function, so we improved and optimized it in conjunction with the ACA.

### 2.2.5. Improved A* Algorithm (Imp-A*)

The A* uses a heuristic function to evaluate the cost from the current node $\varepsilon$ to the target node e. In grid search mapping, the heuristic function refers to the Manhattan distance from the current node to the target node:

$$h(\varepsilon) = |S_{ex} - T_{\varepsilon x}| + |S_{ey} - T_{\varepsilon y}| \tag{9}$$

where $S_{ex}$ and $S_{ey}$ are the x and y coordinates of the current node, and $T_{\varepsilon x}$ and $T_{\varepsilon y}$ are the x and y coordinates of the target node.

Due to the existence of obstacles in the actual environment, $h(\varepsilon)$ in Equation (9) is not necessarily smaller than $h'(\varepsilon)$ in Equation (7); this will result in the shortest route in the plan not necessarily being the best or taking the least time. Based on this question, we introduce the Imp-A*. Imp-A* searches from source O to target grid node e and also searches from target node e to source O. The algorithm terminates when the same grid node is searched in both directions. The optimization of the grid method transplanting path for inserted tray seedlings is a non-negative weighting problem, and there are no other dynamic influences, so this study incorporates ACA to train the heuristic function of A* to improve the efficiency of A*.

The speed of the algorithm depends on the heuristic function, and using ACA to train the heuristic function improves the flexibility of the algorithm and brings the algorithm closer to the actual value. However, the biggest drawback of ACA is its tendency to fall into local optimum solutions. In this paper, we propose an ACA based on parallel rewards for this problem, where parallelism means multiple backups for ants, and rewards mean additional information to the optimal path as well as rewarding the pheromone of the paths adjacent to the optimal path. If additional pheromones are rewarded to the current optimal path, the formula is as follows:

$$\Delta \ell_{ij}^{*}(\varepsilon_j) = \rho' \Delta \ell_{ij}(\varepsilon_j) \tag{10}$$

where $\Delta \ell_{ij}^{*}(\varepsilon_j)$ is the additional reward pheromone, $\rho'$ is the reward factor, and $\varepsilon_j$ is the set of nodes.

Pheromone update formula:

$$\ell_{ij}(\varepsilon_j) = \rho \times \ell_{ij}(\varepsilon_j) + \Delta \ell_{ij}(\varepsilon_j) + \Delta \ell_{ij}^{*}(\varepsilon_j) \tag{11}$$

The pheromone of the reward to the neighboring path is denoted as:

$$\Delta \ell_{ij}^{*}(\varepsilon_j) = \left(\rho'\right)^{m} \cdot \Delta \ell_{ij}(\varepsilon_j) \tag{12}$$

where m is the reward coefficient ($m \in N$) and m is positively related to the distance between adjacent and optimal paths. Assuming that in a grid system of n nodes, m artificial ants are moving at speed $\upsilon_k$. Ant k is searching for the optimal path, and the current node is evaluated using a bidirectional A* based on a parallel reward ant colony system to train the heuristic function. Initialize the pheromone of the raster topology edge as $< \varepsilon_i, \varepsilon_j >$, then:

$$\ell_{ij}(\varepsilon_j) = \text{const} \tag{13}$$

where const is a constant, $\Delta \ell_{ij}(\varepsilon_j) = 0$, then $\ell_{ij}(\varepsilon_j) = 1/d_{ij}$, where $d_{ij}$ is the distance from the node $\varepsilon_i$ to node $\varepsilon_j$, then the following equation is established:

$$\mu_{ij}(\varepsilon_j) = 1 / \left( \frac{\left| TC_{ij}^{max} - TC_{ij}^{design} \right|}{\left| TC_{ij}^{average} - TC_{ij}^{design} \right|} \cdot \frac{L_{ij}^2 \rho_{ij}}{\upsilon_k} \right) \tag{14}$$

where $\mu_{ij}(\varepsilon_j)$ is the heuristic information function, $TC_{ij}^{max}$ is the hourly maximum ant colony flow, $TC_{ij}^{design}$ is the set hourly ant colony flow, $TC_{ij}^{average}$ is the hourly average ant colony flow, $\upsilon_k$ is the current velocity, $\rho_{ij}$ is the current flow density, and $L_{ij}$ is the path distance. $N_{max}$ is the maximum number of cycles of the ACA, which is achieved by the following steps.

Assuming that $\varepsilon_O$ is the source node and $\varepsilon_e$ is the target node, the grid map is searched from the original node using the bidirectional A* algorithm. If the node $\varepsilon_i$ is not the target node, it needs to be extended in the next cycle, denoting the set $\varepsilon_j$ of this node denoted as:

$$f'(\varepsilon_j) = g(\varepsilon_j) + h'(\varepsilon_j) \tag{15}$$

Using a parallel reward ant colony system to train $h'(\varepsilon_j)$, computing from the current node $\varepsilon_{jx}$ to the target node $\varepsilon_{jx} \in \varepsilon_j$, placing m ants and their k backup ants into the node $\varepsilon_{jx}$, using the following equation for selection:

$$p_{ij}^k(V_{jx}) = \begin{cases} \arg\max\left\{\left[\ell_{ij_x}^k(\varepsilon_{j_x})\right]^\alpha \left[\eta_{ij_x}^k(\varepsilon_{j_x})\right]^\beta\right\}\cdots(1) & q \le q_0 \\ \dfrac{\left[\ell_{ij_x}^k(\varepsilon_{j_x})\right]^\alpha \left[\eta_{ij_x}^k(\varepsilon_{j_x})\right]^\beta \left[\mu_{ij_x}^k(\varepsilon_{j_x})\right]^\gamma}{\sum_{x=1}^n \left[\ell_{ij_x}^k(\varepsilon_{j_x})\right]^\alpha \left[\eta_{ij_x}^k(\varepsilon_{j_x})\right]^\beta \left[\mu_{ij_x}^k(\varepsilon_{j_x})\right]^\gamma}\cdots(2) & \text{otherwise} \end{cases} \tag{16}$$

where $q_0$ is a fixed value and q is a random number. The next node is selected according to Equation (1) if $q_0 \ge q$; otherwise, the next node is selected according to Equation (2).

Each ant will choose a least-cost path after one cycle, and the pheromone will be updated as the ant chooses its path:

$$\ell_{ij}(\varepsilon_j) = (1-\rho)\ell_{ij}(\varepsilon_j) + \Delta\ell_{ij}^k(\varepsilon_j) \tag{17}$$

where $\rho$ is the pheromone volatility and $\Delta\ell_{ij}^k(\varepsilon_j) = Q/d_{ij}$ is the total amount of pheromone released by ant k on the pathway.

When m ants have all reached the target node, the optimal path is selected based on the minimum cost path obtained by the global pheromone update in step (3), and additional pheromones are rewarded according to the following equation:

$$\ell_{ij}(\varepsilon_j) = (1-\rho)\ell_{ij}(\varepsilon_j) + \Delta\ell_{ij}(\varepsilon_j) + \Delta\ell_{ij}^*(\varepsilon_j) \tag{18}$$

where $\Delta\ell_{ij}^*(\varepsilon_j) = \rho'\Delta\ell_{ij}(\varepsilon_j)$, $\rho'$ is the pheromone reward factor and:

$$\Delta\ell_{ij} = \sum_{k=1}^m \Delta\ell_{ij}^k \tag{19}$$

where $\Delta\ell_{ij}$ is the amount of pheromone reward, and are the pheromones released by ant k on the path. $\Delta\ell_{ij}^k = Q/L$, where L is the distance traveled by the ants on the optimal path to reach the target node in one cycle. Moreover, additional pheromones need to be rewarded to neighboring paths:

$$\ell_{ij_y}\left(\varepsilon_{j_y}\right) = (1-\rho)\ell_{ij_y}\left(\varepsilon_{j_y}\right) + \Delta\ell_{ij_y}^*\left(\varepsilon_{j_y}\right) \tag{20}$$

where $\Delta\ell_{ij_y}^*\left(\varepsilon_{j_y}\right) = (\rho')^n\Delta\ell_{ij_x}\left(\varepsilon_{j_x}\right)$, and $\Delta\ell_{ij_y}^*\left(\varepsilon_{j_y}\right)$ are the amount of pheromone reward.

When the number of loops reaches Nmax, the optimal path obtained is compared and the smallest path is selected. The estimated cost of the optimal path is based on the following formula:

$$h(\varepsilon_j) = \text{Min}\sum T \tag{21}$$

where $T = L_{ij}/v_k + p_{ij}T_{ij} + pT$ is the length of each road segment on the optimal path.

To find the best path to the other nodes, get an estimated distance for each node through the following steps:

$$f\left(\varepsilon_{j_{oe}}\right) = g\left(\varepsilon_{j_{oe}}\right) + h\left(\varepsilon_{j_{oe}}\right) \tag{22}$$

where $\varepsilon_{j_{oe}} \in \varepsilon_j$ , insert nodes into an openlist and rank them in descending order according to their assessment.

Storing the source points in the Openlist table, indicating that the source node O is the smallest node, and then set the cost of O to $g_1(o) = 0$ , and the cost of the other nodes to infinity.

After accessing the original node, perform the following actions on the remaining node n:

- Calculate the cost of each successor node $\varepsilon$ using the concurrent reward-based ACA:

$$f'_1(\varepsilon) = g_1(\varepsilon) + h'_1(\varepsilon) \tag{23}$$

- Set visiting rules to give priority to all nodes with status "1" in the S grid and then to those with status "0" in the T grid.
- Search for the target node and add it to the Openlist.

Delete the node m with the smallest cost $f'_1(\varepsilon)$ from the Openlist, and change the search status of the node $\varepsilon$ to Least, determine whether node m satisfies the above two termination conditions. If they are satisfied, the last step is performed; if not, the following steps are performed at the next node.

Determining the termination node k of a bidirectional search based on the above termination conditions can obtain the same optimal path as the A* algorithm but greatly reduces the algorithm's running time. In addition, two backtracking searches were conducted, including two paths from source node O to termination node k and from termination node k to target node e, to complete the calculation of the optimal path. Figure 7 shows a schematic of the bidirectional search principle.

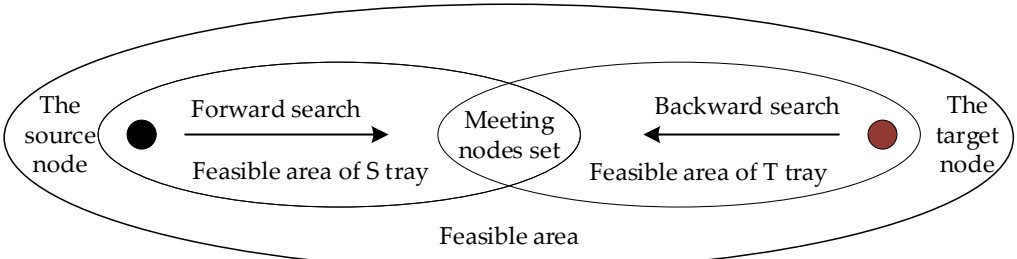

**Figure 7.** Bidirectional A* schematic.

The heuristic function of A* was trained and improved using ACA, and it achieved bi-directional guidance during path finding. When the program starts running, one search path starts from the source point O and visits a healthy seedling grid in the S tray, and the visited healthy seedling grid is added to Closelist and then visits a hole grid in T tray, and the visited hole grid is added to Openlist. While the other search path starts from the source point O and visits a hole grid in T tray, then visits a healthy seedling grid in S tray. Both search paths start at the same time, and the algorithm terminates when both search paths meet at the same node, and all the hole grids in the T tray have been visited. After several searches, the paths of the bidirectional search are connected to the optimal path. Figure 8 shows the detailed search flow diagram of Imp-A*; Figure 9 shows a schematic diagram of the optimal path for path planning using Imp-A* for a 50-hole tray.

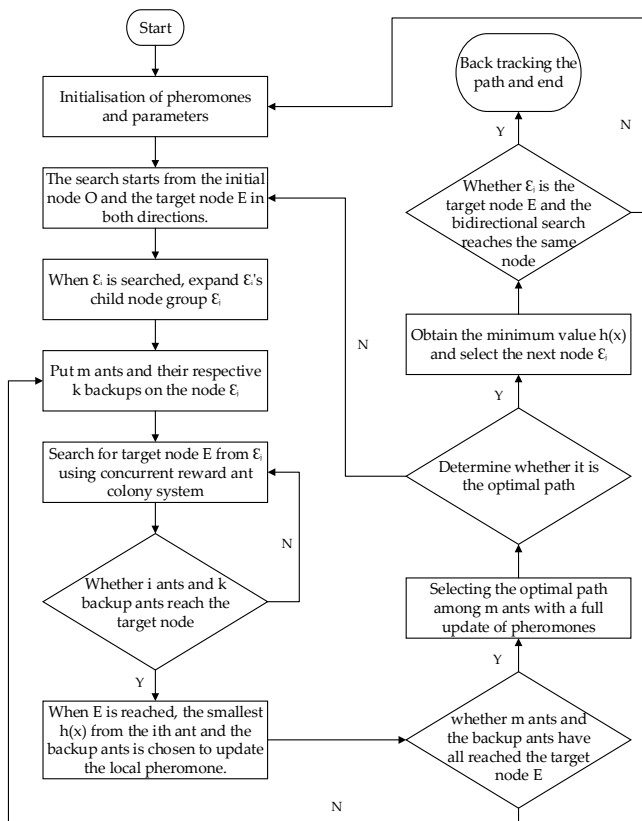

**Figure 8.** Flow chart of the Imp-A algorithm.

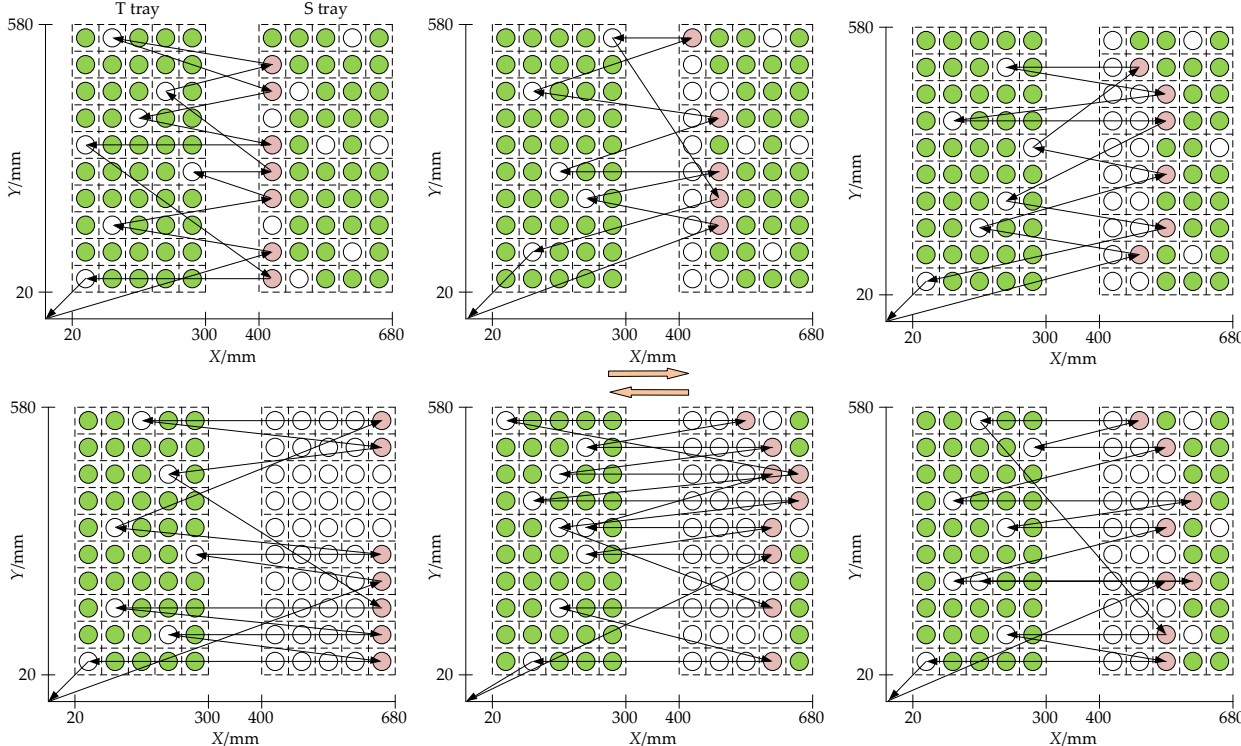

**Figure 9.** Diagram of Imp-A* model to complete 50 holes transplanting path planning. The red circles indicate healthy seedlings to be transplanted; the green circles indicate healthy seedlings; the white circles indicate empty holes.

### 3. Results

To validate the efficacy and performance of the Imp-A* designed in this paper, migration path planning simulations were performed using 32-, 50-, 72-, and 128-hole plug trays. The simulation environment is PYTHON 3.6, and the algorithm compiler is PyCharm Community Edition 2020.3.3.

*3.1. Randomized Comparative Simulation Test*

To verify the advantages and disadvantages between the Imp-A* and other algorithm models, comparative experiments using the CSM, DA, ACA, and A* as comparison schemes with metrics of transplant path planning length and computation time. Since the CSM has a known transplanting route and does not involve computing time, the CSM is only involved in the transplanting path length comparison and not in the algorithm computing time comparison.

Choose from 32, 50, 72, and 128 holes for simulation, with the T tray and S tray the same size. The number and location of no seedling plug were randomized, the task of replanting one destination tray was defined as one section, and the number of healthy seedlings in an S tray was defined as one group. Poor quality seedlings accounted for 5–20% of the plug trays and required an S tray seedling to give 5–10 T trays seedlings. The length of each transplanting path and the time of the algorithm at each node are calculated and summarized as the experimental results of each group. As shown in Table 1, the number of missing seedlings in each group of plug trays for each of the four sizes; for example, in the first set of tests with 32-hole plug trays, the number of healthy seedlings in the S trays was 25, and replanting was carried out to 5 T trays; in the second set of experiments with 32-hole plug trays, the number of healthy seedlings in the S trays was 26, and replanting was carried out to 6 T trays.

**Table 1.** Number of transplanted seedlings for random comparison tests.

| Plug Tray Size/Hole | Test Groups | The Number of T Tray | | | | | | | | | | Total |
|---|---|---|---|---|---|---|---|---|---|---|---|---|
| | | 1 | 2 | 3 | 4 | 5 | 6 | 7 | 8 | 9 | 10 | |
| 32 | 1 | 5 | 6 | 6 | 3 | 5 | - | - | - | - | - | 25 |
| | 2 | 3 | 6 | 4 | 5 | 3 | 5 | - | - | - | - | 26 |
| | 3 | 6 | 3 | 4 | 6 | 5 | 3 | - | - | - | - | 27 |
| | 4 | 2 | 4 | 5 | 5 | 3 | 5 | 4 | - | - | - | 28 |
| | 5 | 4 | 3 | 5 | 6 | 2 | 4 | 5 | - | - | - | 29 |
| | 6 | 5 | 4 | 4 | 3 | 6 | 5 | 3 | - | - | - | 30 |
| 50 | 1 | 7 | 6 | 3 | 5 | 4 | 9 | 6 | - | - | - | 40 |
| | 2 | 5 | 4 | 4 | 7 | 6 | 3 | 8 | 4 | - | - | 41 |
| | 3 | 7 | 5 | 6 | 8 | 9 | 7 | - | - | - | - | 42 |
| | 4 | 3 | 3 | 7 | 6 | 4 | 9 | 5 | 7 | - | - | 44 |
| | 5 | 10 | 6 | 4 | 4 | 7 | 5 | 7 | 3 | - | - | 46 |
| | 6 | 6 | 9 | 7 | 4 | 5 | 7 | 3 | 6 | - | - | 47 |
| 72 | 1 | 8 | 5 | 6 | 4 | 11 | 7 | 9 | 5 | - | - | 55 |
| | 2 | 11 | 7 | 8 | 8 | 5 | 6 | 6 | 7 | - | - | 58 |
| | 3 | 9 | 5 | 7 | 8 | 7 | 6 | 10 | 8 | - | - | 60 |
| | 4 | 7 | 5 | 5 | 6 | 8 | 10 | 6 | 4 | 10 | - | 63 |
| | 5 | 10 | 8 | 10 | 5 | 6 | 7 | 8 | 7 | 5 | - | 65 |
| | 6 | 6 | 11 | 9 | 10 | 9 | 5 | 7 | 11 | - | - | 68 |
| 128 | 1 | 12 | 9 | 15 | 11 | 8 | 10 | 9 | 13 | 15 | - | 102 |
| | 2 | 16 | 10 | 12 | 9 | 13 | 9 | 11 | 10 | 16 | - | 106 |
| | 3 | 11 | 14 | 10 | 9 | 15 | 12 | 14 | 8 | 9 | 9 | 111 |
| | 4 | 15 | 9 | 12 | 11 | 9 | 12 | 13 | 11 | 14 | 9 | 114 |
| | 5 | 12 | 14 | 8 | 19 | 10 | 11 | 9 | 12 | 8 | 15 | 118 |
| | 6 | 18 | 11 | 9 | 9 | 10 | 13 | 8 | 12 | 17 | 14 | 121 |

## 3.2. Analysis of Randomized Comparative Simulation Test Results

Figure 10 shows the results of the 32-, 50-, 72-, and 128-hole path planning length simulations. DA, A*, and Imp-A* have a similar length of complementary path planning, significantly shorter than CSM, and ACA is superior to the CSM in the length of pathway planning but not as long as DA, A*, and Imp-A*. Based on the results, the statistical transplanting path planning lengths for CMS, DA, ACA, A*, and Imp-A* in the 32-hole plug trays simulation test were 208,690.0, 176,120.0, 193,250.0, 16,560.0, and 163,10.0 mm, respectively; 50-hole path planning lengths were 289,238.0, 269,252.0, 276,670.0, 261,354.0, and 25,666.0 mm, respectively; 72-hole path planning lengths were 428,337.0, 315,354.9, 412,227.0, 305,276.4, and 303,399.0 mm, respectively; 128-hole path planning lengths were 740,250.0, 599,274.5, 701,215.0, 584,310.0, and 580,490.0 mm, respectively. By comparison, we can see the superiority of the five path planning algorithm models in the following order: Imp-A*, A*, DA, ACA, and CSM. Imp-A*, A*, and DA perform significantly better than other algorithm models in terms of path planning length when plugs were increased to 72 and 128 holes. When the number of holes increases, the invalid search of the ACA increases sharply and is more likely to fall into the local optimum, the Imp-A* has a precise heuristic function of bidirectional guidance, and it can find the optimal path more precisely.

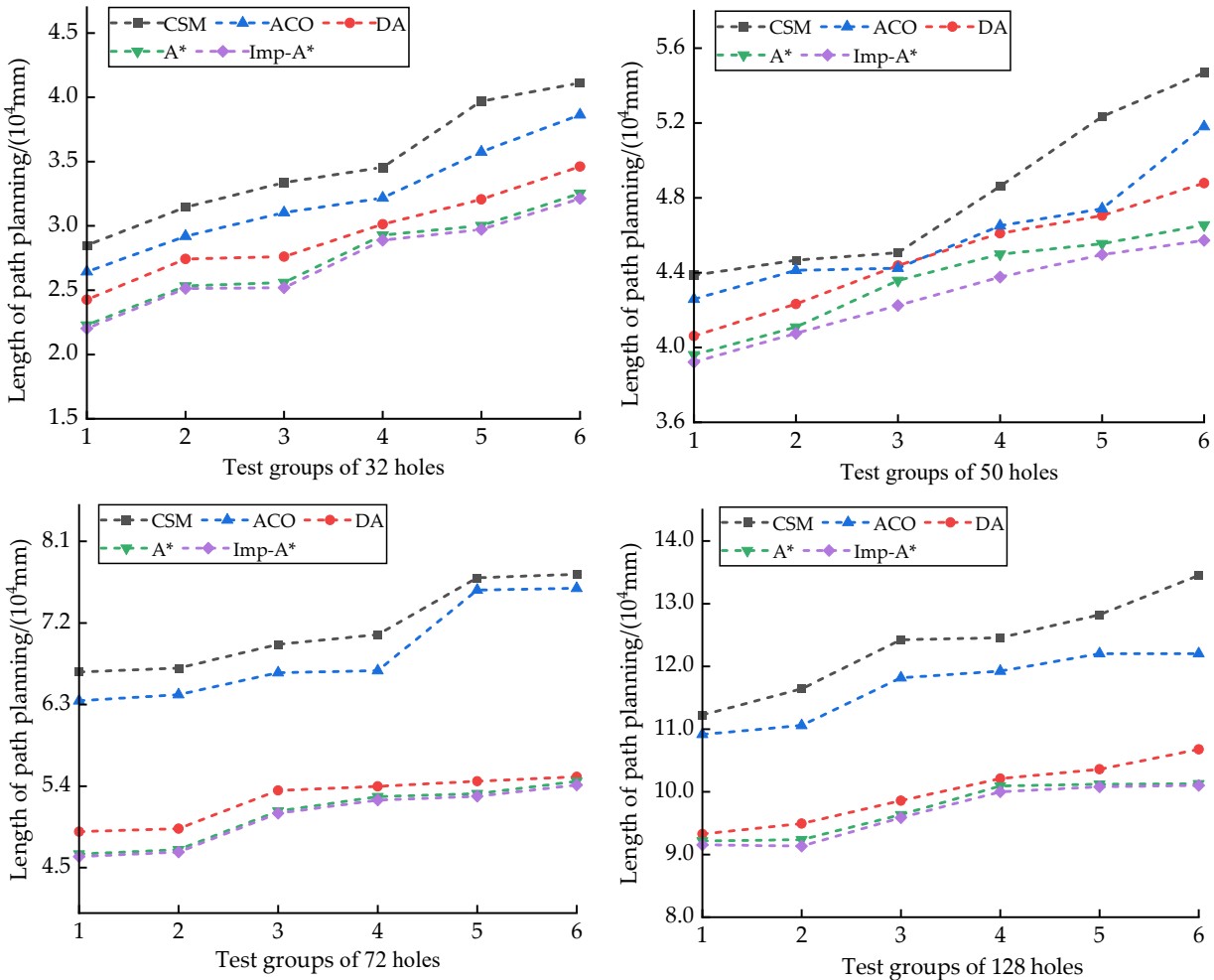

**Figure 10.** Comparison of planned path length for different size plug trays based on simulation tests.

To demonstrate the superiority of Imp-A* in transplantation path planning, the differences between Imp-A* and the other four transplantation path planning length ratio algorithm models were calculated, as shown in Figure 11. At 32 holes, the optimal path planning length for Imp-A* was 25.1% for CSM, 9.2% for DA, 18.8% for ACA, and 1.6% for

A*. At 50 holes, Imp-A* had a 16.4% better path planning length than CSM, 6.3% better than DA, 11.7% better than ACA, and 3.1% better than A*. The maximum path planning length for Imp-A* was 16.4% for CSM, 6.3% for DA, 11.7% for ACA, and 3.1% for A*. Imp-A* had a maximum path planning length of 31.3% compared to CSM, 5.6% compared to DA, and 29.7% compared to ACA under 72 holes. At 128 points, the maximum path planning length for Imp-A* was 24.9% for CSM, 5.4% for DA, 18.9% for ACA, and 1.1% for A*. Data show that Imp-A* has a significantly shorter path planning length than CSM and ACA and a significantly shorter path planning length than DA.

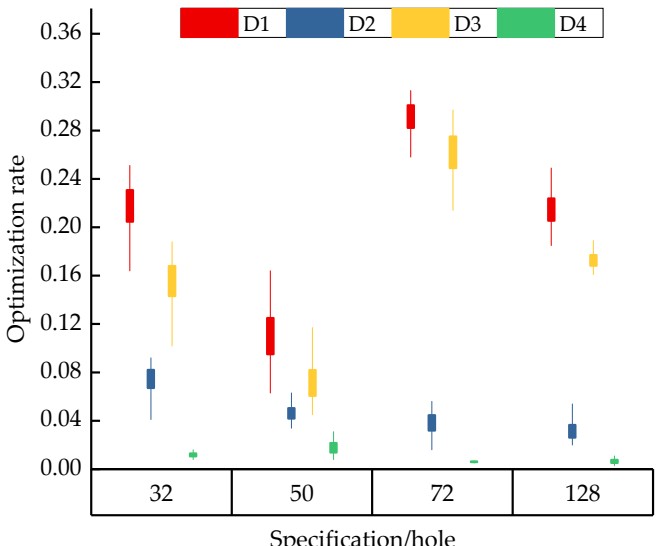

**Figure 11.** Changes in the path planning length difference ratio between Imp-A* and other algorithmic models. D1 = (CSM-Imp-A*)/CSM, D2 = (DA-Imp-A*)/DA, D3 = (ACA-Imp-A*)/ACA, D4 = (A*-Imp-A*)/A*.

Figure 12 shows a comparison of the computation times for different algorithmic models for path planning. It can be seen that the path planning time of A* and Imp-A* is significantly shortened compared with DA and ACA, and the path planning time of Imp-A* is substantially shortened compared with ACA and A* after the improvement of the heuristic function of A* by using ACA. Based on the results of the experiment, the statistical results show that DA, ACA, A*, and Imp-A* took 29.62 s, 19.84 s, 7.97 s, and 4.86 s for 32 holes; 40.73 s, 29.17 s, 11.12 s, and 9.47 s for 50 holes; 56.36 s, 39.84 s, 27.43 s, and 16.46 s for 72 holes; 96.74 s, 86.40 s, 48.64 s, and 36.88 s for 128 holes transplanting path, respectively. Comparing four algorithm models, the order of time required for path planning is Imp-A*, A*, ACA, and DA. Because when the number of holes increases, ACA and DA search for invalid nodes more frequently and are more likely to fall into local optimality, whereas Imp-A* has a heuristic function of fast bidirectional guidance to find the optimal path in a shorter time.

To further illustrate the superiorities of Imp-A* in computation time for transplantation path planning, a graph of computational time variation between Imp-A* and three other algorithm models was obtained, as shown in Figure 13. Imp-A* had a maximum calculation time of 86.0% of DA, 80.1% of ACA, and 46.5% of A* in 32 holes. At 50 holes, Imp-A* had a maximum calculation time of 79.9% of DA, 71.5% of ACA, and 20.5% of A*. In 72 holes, the calculation time of Imp-A* was 71.8% compared to DA, 63.0% compared to ACA, and 41.4% compared to A*. At 128 holes, the calculation time of Imp-A* was 63.4% compared to DA and 41.4% compared to A*. In 128 cavities, the calculation time of Imp-A* was optimized by 63.4% compared to DA, up to 58.9% compared to ACA, and 28.1% compared to A*.

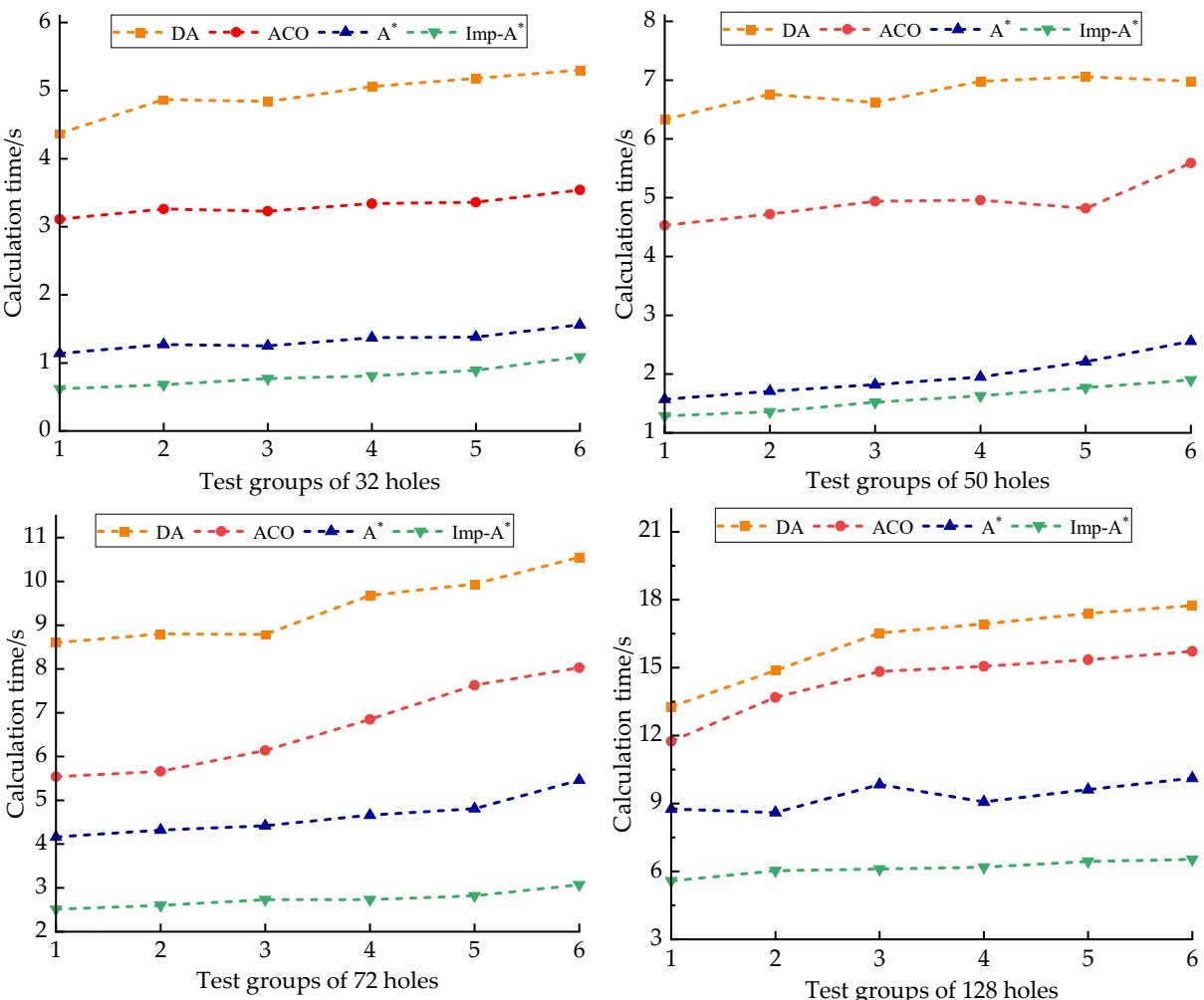

**Figure 12.** Comparison of calculation time for different size plug trays.

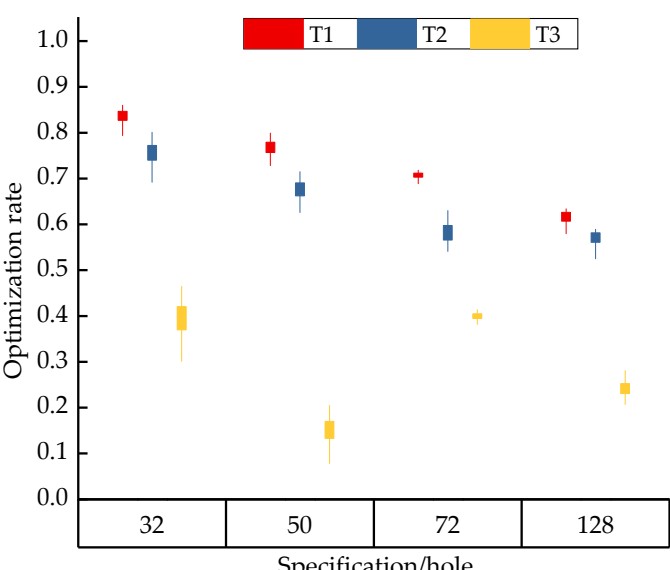

**Figure 13.** Variation of the calculation time difference ratio between Imp-A* and other algorithmic models. T1 = (DA-Imp-A*)/DA, T2 = (ACA-Imp-A*)/ACA, T3 = (A*-Imp-A*)/A*.

As shown in Figure 14a, the standard deviation values of the simulated path planning lengths of the four algorithm models are analyzed. The standard deviation DA, ACA, A*, and Imp-A* path planning lengths were from 1333.44 to 2120.31, from 1264.09 to 2274.91, from 1273.91 to 2169.12, and from 1073.91 to 1869.12, respectively. Figure 14b shows the standard deviation values of the simulated computation times of the four algorithm models. The computation time standard deviation was from 0.16 to 0.33, from 0.17 to 0.27, from 0.12 to 0.20, and from 0.10 to 0.17 for the four algorithm models, respectively. It can be seen that the maximum value of path planning length and computation time Imp-A* are smaller than other algorithm models, and the minimum value is smaller than other algorithm models. The results show that Imp-A* is more stable in obtaining the optimal transplanting path for transplanted seedlings of different specifications than DA, ACA, and A *, with a smaller time gap calculated by algorithmic models.

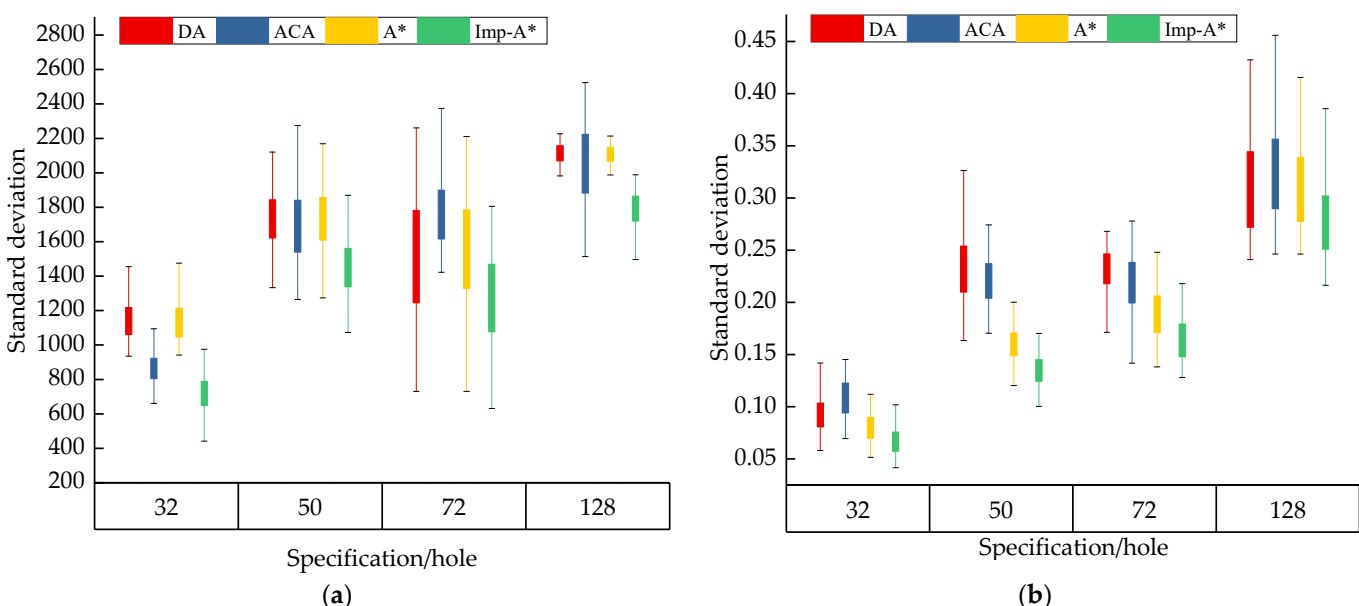

**Figure 14.** Standard deviation diagram of (**a**) path planning length and (**b**) calculation time.

Simulation tests show that Imp-A* has plug trays significant advantage over CSM and ACA and is shorter than DA and A* in the planning length of the transplanting pathway. Imp-A* is significantly shorter than DA and ACA and shorter than A* in terms of computation time in the algorithm model. Imp-A* shows better optimization capability and efficiency, both in terms of path planning length and calculation time.

*3.3. Test Verification*

To verify the performance of the Imp-A, the implants under different algorithmic models were experimentally investigated. The path planning lengths and computation times of CSM, DA, ACA, A*, and Imp-A* for transplanting 50 holes tray seedlings were counted. Figure 15a shows the automatic transplanter developed by our team, and Figure 15b shows the sample transplanter used for the experiment.

Tomato seedlings are placed in cuttings at the age of 15 d and are grown in 50 holes at the "Muxi Greenhouse" nursery in Xinjiang, China. In total, 35 trays of seedlings were selected, and 30 of these trays were divided into five groups, with six trays in each group serving as T trays and the number of missing seedlings in each group corresponding to the number of trays in each group. The remaining five trays were used as S trays, and the number of seedlings lost per T tray is shown in Table 2. Before the test, a visual inspection of the plug seedlings was performed, the plug substrate was derived, and unhealthy plug seedlings were removed from the T tray by a manipulator (the lab-developed plug seedling visual inspection device and the transplanter were two parts of the test that

have not incorporated but do not affect the test). At the beginning of the experiment, one path planning algorithm was selected to transplant the first group of 6 T trays, and after completing the task (and all the time it took), another tray was replaced with another until healthy seedlings were transplanted from the S tray. The remaining four groups were then transplanted using other algorithms. The distance between the T tray and S tray is 100 mm, depending on the actual demand. The manipulator is known to move at speeds of 200 mm/s, 200 mm/s, and 100 mm/s in the X, Y, and Z directions, respectively. The manipulator takes to fall by 1.0 s and rise by 0.8 s (the transplanter is more stable). The length of the replanting path (known as the replication path) and time (known as the replanting time) required for the manipulator to complete each plug of seedlings are recorded, as shown in Table 2.

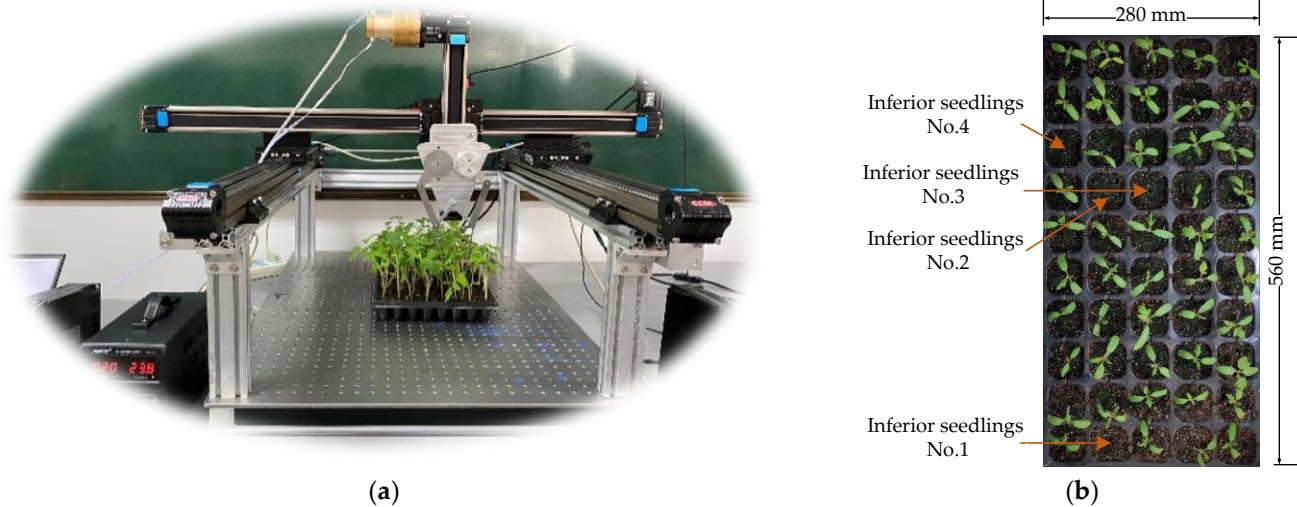

(**a**)  (**b**)

**Figure 15.** Diagram of (**a**) automatic transplanter and (**b**) plug tray seedlings sample.

**Table 2.** Comparison of plug seedling replanting results for different algorithms.

| No. of T Tray | Number of Transplanted Seedlings | Path Planning Length (mm) | | | | | Calculation Time (s) | | | | Replanting Time (s) | | | | |
|---|---|---|---|---|---|---|---|---|---|---|---|---|---|---|---|
| | | CSM | DA | ACA | A* | Imp-A* | DA | ACA | A* | Imp-A* | CSM | DA | ACA | A* | Imp-A* |
| 1 | 3 | 2912.0 | 2686.0 | 2804.0 | 2642.0 | 2596.0 | 0.40 | 0.36 | 0.31 | 0.24 | 36.56 | 35.43 | 36.02 | 32.21 | 30.98 |
| 2 | 4 | 4536.0 | 4262.0 | 4398.0 | 4236.0 | 4210.0 | 0.45 | 0.44 | 0.40 | 0.28 | 44.68 | 43.31 | 43.99 | 39.18 | 38.05 |
| 3 | 6 | 6108.0 | 5230.0 | 5620.0 | 5186.0 | 5186.0 | 0.66 | 0.62 | 0.58 | 0.44 | 52.54 | 48.15 | 50.10 | 41.93 | 40.37 |
| 4 | 7 | 7088.0 | 6770.0 | 6906.0 | 6724.0 | 6708.0 | 0.71 | 0.68 | 0.63 | 0.48 | 77.44 | 75.86 | 76.53 | 68.62 | 67.54 |
| 5 | 9 | 8662.0 | 8466.0 | 8512.0 | 8396.0 | 8372.0 | 1.09 | 1.02 | 0.96 | 0.72 | 95.31 | 94.33 | 94.56 | 85.98 | 84.06 |
| 6 | 10 | 9680.0 | 9024.0 | 9346.0 | 8986.0 | 8924.0 | 1.22 | 1.21 | 1.04 | 0.79 | 120.40 | 117.12 | 118.73 | 106.93 | 105.02 |
| Total | 39 | 38,986.0 | 36,438.0 | 37,586.0 | 36,170.0 | 35,996.0 | 4.53 | 4.33 | 3.92 | 2.95 | 426.93 | 414.20 | 419.93 | 374.85 | 366.02 |
| | Average | 999.6 | 934.3 | 963.7 | 927.4 | 923.0 | 0.12 | 0.11 | 0.10 | 0.08 | 10.95 | 10.62 | 10.77 | 9.61 | 9.39 |

As can be seen from Table 2, CSM, DA, ACA, A*, and Imp-A* have significantly improved performance in practice. The total path planning length of these five algorithm models was 38,986.0, 36,438.0, 37,586.0, 36,170.0, and 35,996.0 mm, respectively, and optimization capability affected: Imp-A* > A* > DA > ACA > CSM. Imp-A*, A*, and DA have similar path planning lengths, 7.67%, 7.22%, and 6.54% shorter than CSM, and 4.23%, 3.78%, and 3.01% shorter than ACA, respectively. The calculation times of DA, ACA, A*, and Imp-A* were 4.53, 4.33, 3.92, and 2.95 s, respectively, and Imp-A* was 34.88%, 31.87%%, and 24.74% less than DA, ACA, and A*, respectively. The total replanting time was 426.93, 414.20, 419.93, 374.85, and 366.02 s using CSM, DA, ACA, A*, and Imp-A* for a total of 39 × 6 holes in the 6 T trays, in descending order: Imp-A*, A*, DA, ACA, and CSM. While the ACA's calculation time is shorter than DA's, its path planning is longer than DA's, leading to longer manipulator execution during the transplant process. Imp-A* saved 60.91 s on CSM, 48.18 s on DA, 53.91 s on ACA, and 8.83 s on A*. The average value of the

length of the transplanting path and the time taken by the manipulator to complete each seedling in a plug tray. It is easy to see that Imp-A* performs best in terms of path planning length, computation time, and replantating time.

Data from six transplantation trials in Table 2 were analyzed. Figure 16 shows the optimization rate of Imp-A* relative to the other four algorithm models in terms of path planning length, computation time, and replanting time. It can be seen that Imp-A* has the highest optimization rate of 40.00% and 36.36% in terms of computation time compared to DA and ACA. Imp-A* has superior performance in computation time and can be applied to a variety of plug tray specifications. Validation tests show that the Imp-A* proposed in this paper performs well and has a good path optimization capability, and the practical results match the simulation results.

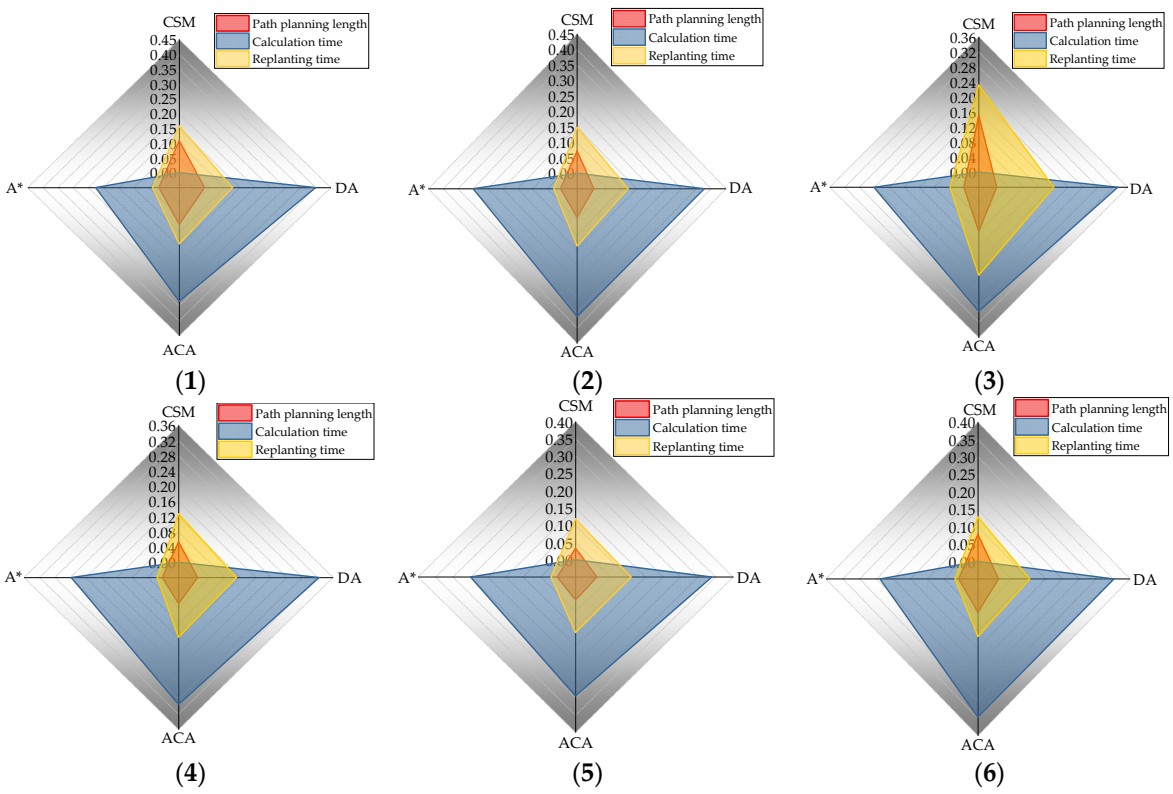

**Figure 16.** Comparison of the optimization efficiency of Imp-A* with other algorithms (**1–6**).

## 4. Discussion

The layout of transplanted seedlings was simulated by establishing a matrix grid of transplanted seedlings. Simulation tests were carried out by using the path optimization model. The path planning length and computation time of different algorithm models were obtained. Simulation results show that the proposed algorithm has good performance in path planning length and algorithm computation time. To verify the reliability and practicality of the algorithm model, experiments were carried out on the transplanting of plug tray seedlings. The results show that the Imp-A* can plan the optimal transplanting path for the manipulator in less time, and the efficiency of the transplanter can be improved by using a superior algorithm model when the transplanter control system is determined.

Imp-A* had an average path planning length of 27,185.0, 43,844.3, 50,733.2, and 96,765.0 mm, and average calculation times of 0.83, 1.58, 2.72, and 6.81 s, respectively, when plug tray size was 32, 50, 72, and 128 holes, respectively, in a simulation test of Section 3.2 of 128 holes of paper. In the transplanting trials in Section 3.3 of this paper, the time taken to transplant Imp-A* was calculated to be 2.4%, 12.84%, 11.63%, and 14.27% shorter compared to A*, ACA, DA, and CSM, respectively. In the literature [23], by analyzing the performance of the algorithm GA and ACA from the perspective of path planning length

and computation time, it was concluded that ACA is superior to GA in path planning length but far less than GA in computation time. Due to the difference in the size of the insert plug tray and the motion of the manipulator, the data of the path planning length proposed in this paper cannot be compared with the literature [23], but it has obvious advantages in computation time. By applying the greedy algorithm (GRA) model to the transplantation path planning problem [17], the path planning length of different holes is obtained, and the data are similar to this paper. It can be seen from the above that although DA's path planning length is similar to that of Imp-A*, the significant difference in algorithm model computation time is because GRA's computation time is limited by local search elements, and the optimal solution cannot be found quickly.

Path planning length and computation time are two important indexes of the algorithm model. The length of path planning and computation time will directly affect transplanting efficiency and actual production efficiency. The Imp-A* algorithm developed in this paper has good performance in both aspects; the algorithm is stable and can provide a reference for the optimization of the migration path. However, the algorithm model in this paper has some shortcomings. For example, this algorithm model is suitable for transplanting seedlings in transplanters of the same specification. If the size of the transplanter and T tray is different, the algorithm model needs to be improved to adapt to different operating environments.

## 5. Conclusions

In this paper, the heuristic function of A* is trained and improved, and the path optimization performance of each algorithm is compared by simulation tests and transplantation trials.

In the 50-hole path planning length simulation test, the average path lengths of Imp-A*, A*, ACA, DA, and CSM to complete the transplanting task in one T tray differed significantly. The ACA did not perform as well in path planning, with Imp-A* and A* performing more prominently as the hole tray sizes became larger. In the 50-hole operation time simulation test, the average running time for Imp-A*, A*, ACA, and DA to complete a T tray transplanting task differed significantly. The DA performed the worst in terms of algorithm calculation time, with the DA and ACA significantly decreasing as the hole size became larger and the Imp-A* still maintaining a fast calculation time. In the 50-hole transplanting trial, the transplanting time of different algorithm models is very different. The actual performance of Imp-A* matches the simulation results, and the path planning algorithm developed in this paper for transplanting seedlings in plug trays performs well and is ideal for practical applications. In mass production, transplanting path planning by the transplanting method can save a lot of time and improve the efficiency of transplanting machinery.

Taken together, Imp-A* inherits the strengths of ACA and A* and has significant advantages in transplant path planning simulation tests and practical production. This research can provide ideas and references for improving the efficiency of transplanting machinery and accelerating the automation and intelligence of transplanting machinery.

**Author Contributions:** Conceptualization, X.L. and W.W.; methodology, X.L. and W.W.; software, X.L. and W.W.: validation, X.L., G.L. and R.L.; formal analysis, F.L.; investigation, X.L.; resources, W.W.; data curation, X.L., writing—original draft preparation, X.L.; writing—review and editing, W.W.; visualization, W.W.; supervision, R.L. and F.L. All authors have read and agreed to the published version of the manuscript.

**Funding:** This research was funded by The National Natural Science Foundation of China (Grant No. 61763042).

**Data Availability Statement:** The datasets used and/or analyzed during the current study are available from the corresponding author on reasonable request.

**Acknowledgments:** The authors gratefully acknowledge the financial support provided by The National Natural Science Foundation of China (61763042). Any opinions, findings, conclusions, or

recommendations expressed in this publication are those of the authors and do not necessarily reflect the view of Shihezi University.

**Conflicts of Interest:** The authors declare no conflict of interest.

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
