# Peer review of "Optimizing the Path of Plug Tray Seedling Transplanting by Using the Improved A* Algorithm"

_agriculture, doi:10.3390/agriculture12091302_

Round 1
Reviewer 1 Report
1- avoid similarities between abstract and conclusion
2- the study is based on A* algorithm(A*), please giving the identity of it in the abstract or at the start of the introduction
3- there is no space between numerous words
4- no statical analysis in the tables
5- in the title by using please delete by ( using only)
Author Response
Author Response 1
Dear peer reviewers and editors
Hello! Thank you very much for your professional and wise comments on this article. According to the experts' questions and opinions, the author gives the detailed explanations in the form of one question and one answer, and marks the manuscript with red font in the corresponding position, as follows:
QUESTION1: Avoid similarities between abstract and conclusion .
RE: Thank you very much for your valuable comments. The authors have rewritten the conclusions of the manuscript in a more general, synthetic manner and have marked the rewritten sections in red to avoid similarities between the conclusions and the abstract.
QUESTION2: The study is based on A* algorithm(A*), please giving the identity of it in the abstract or at the start of the introduction.
RE: Thank you very much for your valuable comments. The identity of the A* algorithm is described by the authors in the abstract of the manuscript and marked them in red.
QUESTION3: There is no space between numerous words
RE: Thank you very much for your valuable comments. The author double-checked the missing spaces in the article and made corrections and marked them in red.
QUESTION4: No statical analysis in the tables.
RE: Thank you very much for your valuable comments. The authors have performed a static analysis of the means for the data in Table 2 and have marked them in red.
QUESTION5: In the title by using please delete by ( using only)
RE: Thank you very much for your valuable comments. The author has delete "by" from the title of the manuscript and marked them in red.
Thank you very much for your valuable opinions. Please review the revised article!

Reviewer 2 Report
The manuscript presents a complete study with considerable application potential. The areas concerned are not only agriculture, but also horticultural production, floriculture and forestry in the field of nursery. The course of the research along with the methodological description is transparent. The results seem promising.
Detailed comments:
(1) Keywords should not duplicate the title of the manuscript.
(2) Throughout the publication (including figures and tables), units should be written exponentially and placed in square brackets.
(3) The designation of the mathematical formula number 3 should be on the right side.
(4) In fig. 16, it is better to include one general, legible legend than to repeat it six times.
(5) The initial sentence of the chapter "Discussion" is redundant - the reader knows what the research work was about.
(6) Requests must be redrafted. They should be more synthetic. Currently, they are actually a repetition of research results.
Author Response
Author Response 2
Dear peer reviewers and editors
Hello! Thank you very much for your professional and wise comments on this article. According to the experts' questions and opinions, the author gives the detailed explanations in the form of one question and one answer, and marks the manuscript with red font in the corresponding position, as follows:
QUESTION1: Keywords should not duplicate the title of the manuscript.
RE: Thank you very much for your valuable comments. The author has reworked the keywords to avoid duplication of keywords with the manuscript title and marked it in red.
QUESTION2: Throughout the publication (including figures and tables), units should be written exponentially and placed in square brackets.
RE: Thank you very much for your valuable comments. The authors have carefully checked the entire manuscript (including figures and tables) and have placed the units involved in square brackets in the form of indices and marked it in red.
QUESTION3: The designation of the mathematical formula number 3 should be on the right side.
RE: The author has set the numbering of equation 3 to the right and marked it in red.
QUESTION4: In fig. 16, it is better to include one general, legible legend than to repeat it six times.
RE: Thank you very much for your valuable comments. The authors have re-examined the significance of the existence of Figure 16 and found that the six graphs in Figure 16 are an analysis of the six sets of experimental data in Table 2 and that six graphs are in fact needed to represent them. However, we have labelled the 6 graphs in Figure 16 to facilitate a more intuitive understanding by the reader and marked them in red.
QUESTION5: The initial sentence of the chapter "Discussion" is redundant - the reader knows what the research work was about.
RE: Thank you very much for your valuable comments. The author deleted the first sentence of the "Discussion" section of the manuscript and marked them in red.
QUESTION6: Requests must be redrafted. They should be more synthetic. Currently, they are actually a repetition of research results.
RE: Thank you very much for your valuable comments. The authors have redrafted more synthetic, clearer requesta and avoided its duplication with the findings and marked them in red.
Thank you very much for your valuable opinions. Please review the revised article!
